# Efficacy and safety of paravertebral block versus intercostal nerve block in thoracic surgery and breast surgery: A systematic review and meta-analysis

**Sheng Huan**[1], **Youming Deng**[2], **Jia Wang**[2], **Yihao Ji**[1,3], **Guoping Yin**[2]*

**1** Nanjing University of Chinese Medicine, Nanjing, Jangsu, China, **2** Department of Anesthesiology, The Second Hospital of Nanjing, Nanjing, Jangsu, China, **3** Department of Critical Medicine, The Second Hospital of Nanjing, Nanjing, Jangsu, China

* yinguoping0304@163.com (GY)

**Data Availability Statement:** All relevant data are within the manuscript and its Supporting Information files.

## Abstract

### Objective

To evaluate the analgesic efficacy and safety of paravertebral block (PVB) versus intercostal nerve block (INB) in thoracic surgery and breast surgery.

### Methods

The PubMed, Web of Science, Embase and the Cochrane Library were searched up to February 2020 for all available randomized controlled trials (RCTs) that evaluated the analgesic efficacy and safety of PVB compared with INB after thoracic surgery and breast surgery. For binary variables, odds ratio (OR) and 95% confidence interval (CI) was used. For continuous variables, weighted mean difference (WMD) and 95% confidence interval (CI) were used. RevMan5. 3 and Stata/MP 14.0 were used for performing the meta-analysis.

### Results

A total of 9 trials including 440 patients (PVB block:222 patients; INB: 218 patients) met the inclusion criteria. In the primary outcome, there was no significant differences between the two groups with respect to postoperative visual analogue scale (VAS) at 1h (Std. MD = -0. 20; 95% CI = -1. 11to 0. 71; P = 0. 66), 2h (Std. MD = -0. 71; 95% CI = -2. 32to 0. 91; P = 0. 39), 24h (Std. MD = -0. 36; 95% CI = -0. 73 to -0. 00; P = 0. 05) and 48h (Std. MD = -0. 04; 95% CI = -0. 20 to 0. 11; P = 0. 57). However, there was significant difference in VAS of non Chinese subgroup at 1h (Std. MD = 0. 33; 95% CI = 0. 25to 0. 41; P<0. 00001) and VAS of Chinese subgroup at 24h (Std. MD = -0.32; 95% CI = -0.49 to-0.14; P = 0.0003). In the secondary outcome, the analysis also showed no significant difference between the groups according to the rates of postoperative nausea and vomit (OR = 0. 63; 95% CI = 0. 38 to 1. 03; P = 0. 06) and the rates of postoperative additional analgesia (OR = 0. 57; 95% CI = 0. 21 to 1. 55; P = 0. 27). There was significant difference in postoperative consumption of morphine (Std. MD = -14. 57; 95% CI = -26. 63 to -0.25; P = 0. 02).

**Funding:** The author(s) received no specific funding for this work.

**Competing interests:** The authors have declared that no competing interests exist.

## Conclusion

Compared with INB, PVB can provide better analgesia efficacy and cause lower consumption of morphine after thoracic surgery and breast surgery.

## Introduction

No matter thoracic surgery or breast surgery, postoperative analgesia has always been the focus of attention of anesthesiologists. Postoperative pain without good treatment may cause complications such as unhealing wound, respiratory inhibition, hemodynamic disorder, anxiety and fidgety, leading to prolonged stay of hospital and difficulty of recovery of patients [1–3]. PVB and INB have been popular methods of postoperative analgesia in the recent years because of the popular use of ultrasound [4]. Compared with the traditional standard of epidural analgesia, it has the advantages of less effect on respiratory or circulation system and less postoperative complications such as epidural hematomas and irreversible neurological disorder [5–6], which can also be applied to patients with coagulation dysfunction.

On the one hand, PVB has a long history. The first PVB was performed by Hugo sellheim in 1905 [7]. thoracic paravertebral space is a potential cavity on both sides of the vertebral body, which is wedge-shaped. The medial wall is composed of vertebral body, intervertebral disc and outer opening of intervertebral foramen; the anterior wall is composed of parietal pleura and intrathoracic fascia; the posterior wall is composed of rib, intercostal muscle and intercostal fascia, transverse process and superior costal transverse process ligament [8]. Each thoracic paravertebral space is not directly connected, but when injected in the near middle part, the local anesthetics can spread cranially and caudally through loose connective tissue [9]. A minority of the injected local anaesthetic can spread laterally to the intercostal space and to the epidural space [10].

On the other hand, INB is also used by many anesthesiologists for postoperative analgesia in thoracic surgery. Anatomically, The intercostal space is the continuation of thoracic paravertebral space to the outside. The intercostal nerve and its accompanying intercostal artery and vein run between the intercostal innermost muscle and the intercostal innermost muscle along the lower edge of the rib. Posterior to the angle of rib, intercostal nerve divided into collateral and lateral cutaneous branches which run into the intercostal muscles and was divided into anterior and posterior branches [11, 12]. The location signs of INB under ultrasound were intercostal muscle, intercostal innermost muscle and pleura.

More and more published studies compared PVB with INB in terms of the VAS and complications [13–21]. But these articles were not well-integrated and the results of them have been found contradictory and unconvincing. For example, some of these studies thought the analgesic effect of PVB is similar with INB but the other did not think so. We had therefore conducted a systematic review and meta-analysis to assess the efficacy and safety of PVB and INB.

## Methods

The meta-analysis was performed in accordance with the Preferred Reporting Items for Systematic Reviews and Meta-Analyses (PRISMA) statement issued in 2009 [22].

### Search strategy

We searched PubMed, Web of Science databases, Embase and the Cochrane Library (up to February 9, 2020) to identify relevant articles. The search terms included the following:

"paravertebral block", "intercostal nerve block", "thoracic surgery", "thoracoscopic surgery" "thoracotomy" "mastectomy" "breast surgery". Appropriate adjustments were made when searching the database. We also checked the references of all the included articles to identify additional relevant studies. There was no restriction of language.

## Inclusion criteria

The inclusion criteria of the study was listed as follows: the study design was RCT the subjects were patients undergoing thoracotomy surgery or thoracoscopic surgery or breast surgery. the comparison was between PVB and INB the conclusion of the study included at least one of the primary or secondary outcomes mentioned below.

## Exclusion criteria

The exclusion criteria of the study was listed as follows: the types of articles were review, case report, experiment of animal, comments, letter and vitro studies. the type of the surgery was not thoracic surgery or breast surgery PVB or INB was not mentioned.

When there were several articles with similar contents and data, we chose the article with most detailed information for our analysis. No limitation of minimum sample sizes were set for exclusion criteria in this analysis. The debate were resolved by discussion of authors.

## Data extraction

The characteristics and outcomes from each included study were checked and extracted by two reviewers (Sheng Huan and Yihao Ji) independently as follows: last name of the first author, publication year, number of patients, age, body mass index, American Society of Anesthesiologists (ASA, I/II/III), operation, intervention groups, details of the interventions.

The primary outcome was VAS at rest recorded 1, 2, 12, 24 hours after surgery. The secondary outcomes included rate of postoperative nausea and vomiting (PONV), rate of additional analgesia and the postoperative consumption of morphine.

## Quality assessment

Two authors (Sheng Huan and Yihao Ji) independently completed the assessment of the quality of reviewed studies according to the Cochrane Collaboration Risk of Bias Tool for randomized controlled trials [23]. Disagreement or discrepancies were resolved by discussion with the third author (Guoping Yin). Quality assessment of 9 included studies was shown in Fig 1.

## Statistical treatment

We used RevMan software (version 5. 3. 5) and Stata/MP (version 14.0) to perform this meta-analysis. For binary variables, odds ratio (OR) and 95% confidence interval (CI) were used. For continuous variables, weighted mean difference (WMD) and 95% confidence interval (CI) were used. Statistical heterogeneity was estimated using the $I^2$ test, which was considered present when $I^2 > 50\%$ or $P < 0.05$, and then and the fixed-effects model was used. If not, the randomized-effects model was used and we then performed sensitivity analysis and subgroup analysis to find out the sources of heterogeneity. Publication bias was performed for every analysis using Egger's test which is shown in the additional file named as Supplementary Information 1. When the P value $< 0.05$, the difference between groups was considered statistically significant.

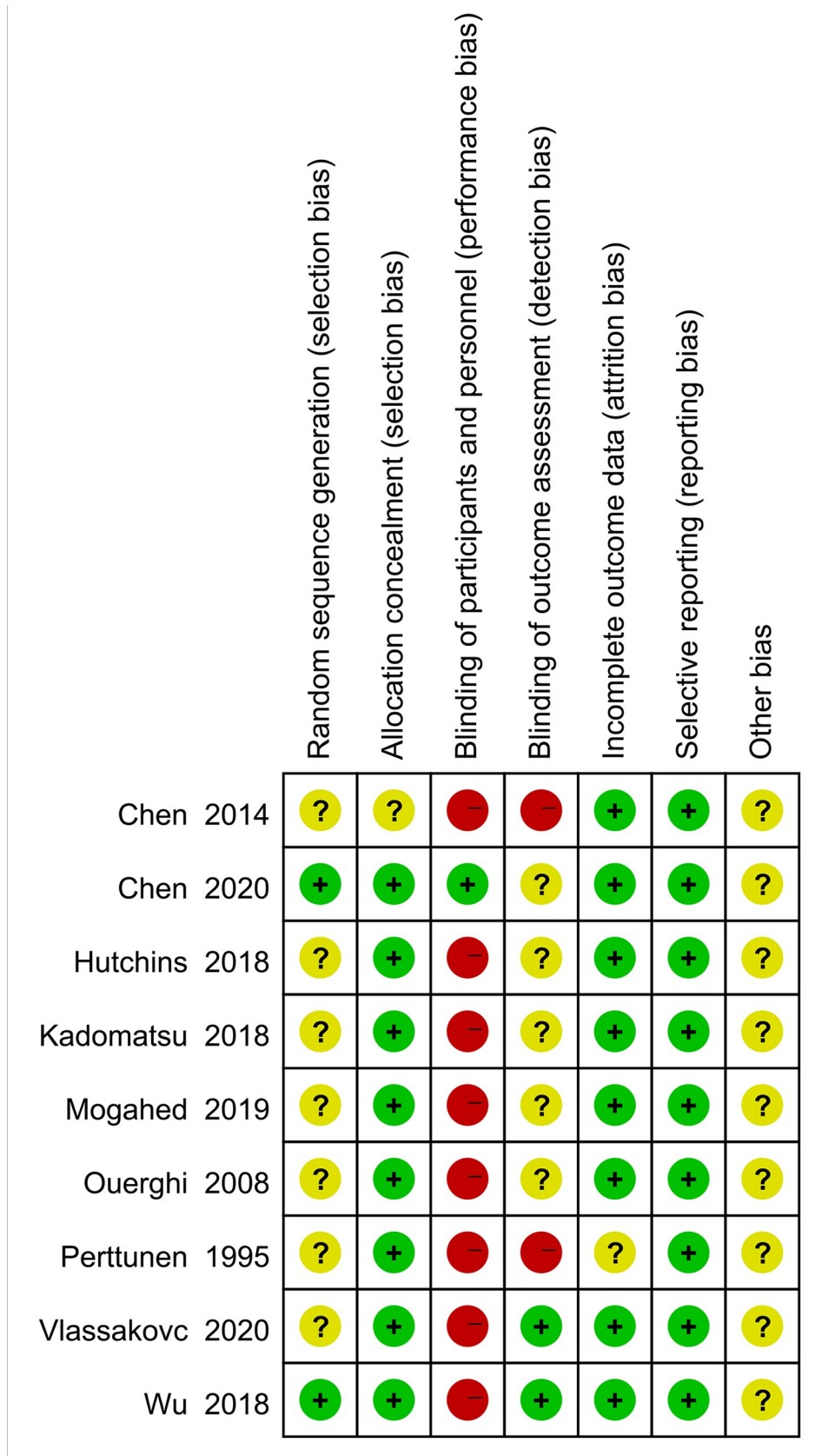

**Fig 1. Risk of bias summary for the included studies.**

After finishing the whole article, it was reviewed and examined again by the two authors (Youming Deng and Jia Wang), who put forward many valuable suggestions. Then one author (Sheng Huan) made further modifications to the article.

## Results

### Identification of eligible studies characteristics of the studies

A total of 613 records were identified by the initial search. After removing the duplicates, there were still 367 unique articles remaining. Then, we reviewed all the remaining abstracts finding that only 18 studies met the inclusion criteria. One of these studies was ruled out due to incomplete data and 6 of them was excluded because of absence of full text. In addition, two studies which are not RCTs were also excluded. Finally, the remaining 9 studies [13–21] were included in our systematic review and meta-analysis. The strategy of the research and the process of the selection were shown in the flow diagram of Fig 2.

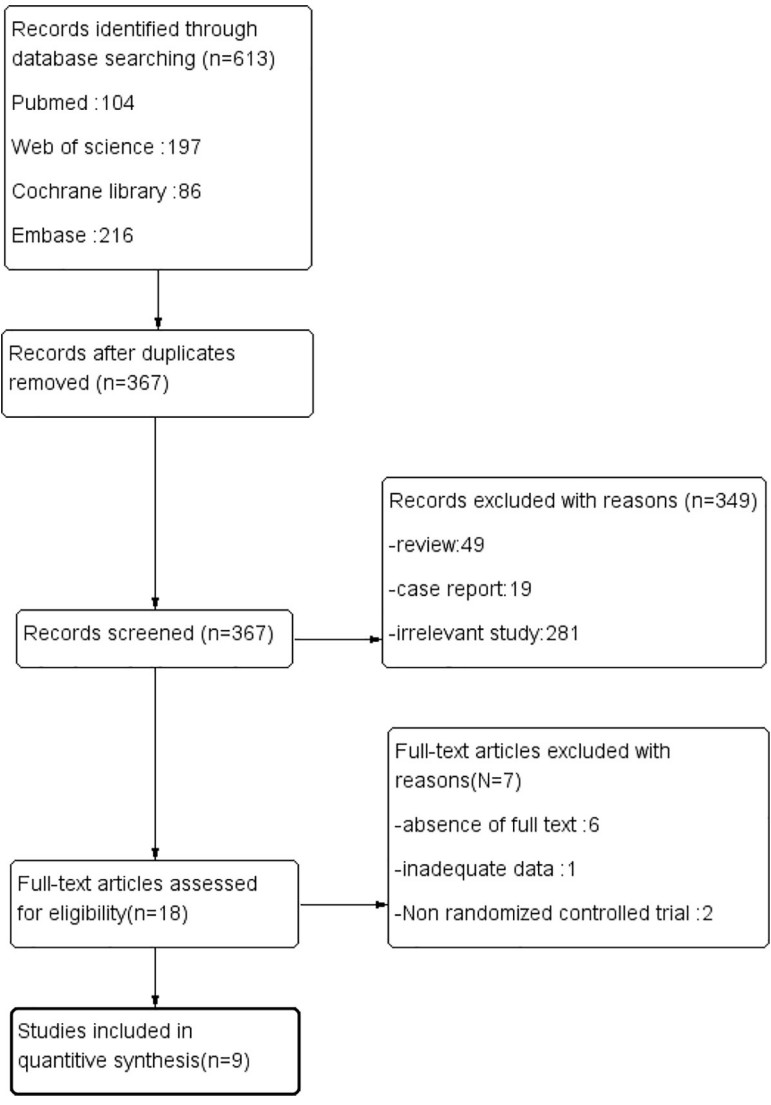

**Fig 2. The search, inclusion and exclusion of the studies.**

## Characteristics of the studies

The characteristics of the studies included were shown in Table 1. All the included trials were published from 1995 to 2020 which enrolled a total of 440 patients, including 222 patients in the PVB group and 218 in the INB group. In some included studies, we cannot contact with the authors to get access to the primary data and we have therefore used some software tools to get the relevant data. The data of VAS at rest of study Perttunen (1995) [21] was presented with the form of mean, maximum and minimum values instead of mean and standard deviation. Therefore, we referred to a statistical method to estimate the variance data [24]. Two studies Mogahed (2019) and Wu (2018) [15, 16] presented the mean and standard deviation of VAS at rest in the form of graphs, so we analysed it with a digitizing software named Engauge Digitizer to get the data. The detailed characteristics of the included studies (9 RCTs) were presented in the file named as Supplementary Information 1.

## Publication bias

We evaluated the publication bias using Egger's test and found that there is no publication bias existing in our meta-analysis. The result was shown in the additional file named as S1 Appendix.

## Primary outcome

**Primary outcome: VAS at rest at the first 1 h.**   Four trials [15, 16, 19, 21] reported the VAS at rest at the first 1 h and we used the random-effect model to analysis the outcome of them. The results showed that compared to INB, PVB resulted in no significantly difference in VAS at rest at the first 1 h (Std. MD = -0. 20; 95% CI = -1. 11to 0. 71; P = 0. 66) with significant heterogeneity among the studies ($I^2$ = 98%, heterogeneity P<0.00001) (Fig 3).

The sensitivity analyses conducted by us found that after omitting one study by turns, significant heterogeneity was still present among the trials. Then we perform the subgroup analysis to investigate that whether the VAS at rest at the first 1 h was influenced by the subjects of different countries. In the 4 trials, subjects of 2 studies Wu (2018) and Chen (2014) [16, 19] were Chinese and they were classified in the subgroup 1 showing no significantly difference in VAS at rest at the first 1 h (Std. MD = -0. 75; 95% CI = -1.83 to 0. 34; P = 0.18) with significant heterogeneity among the studies ($I^2$ = 97%, heterogeneity P<. 00001). At the same time, the other two studies Mogahed (2019) and Perttunen (1995) [15, 21] both including non-Chinese patients was classified in subgroup2, which showed a significantly decrease in VAS at rest at the first 1 h in people accepting PVB (Std. MD = 0. 33; 95% CI = 0. 25to 0. 41; P<0. 00001) with no heterogeneity among the studies ($I^2$ = 0%, heterogeneity P = 0. 51) (Fig 4).

We also performed the subgroup analysis to investigate that whether the VAS at rest at the first 1 h was influenced by the use of Patient controlled analgesia (PCA) consisting of opioid drug. However, after excluding the study Mogahed (2019) [15], the heterogeneity was still very significant ($I^2$ = 98%, heterogeneity P<0.00001) and there is no significant difference between PVB and INB (Std. MD = 0. 45; 95% CI = -1.46to 0. 56; P = 0.38) (Fig 5).

**Primary outcome: VAS at rest at the first 2 h.**   Four trials [16, 17, 19, 21] reported the VAS at rest at the first 1 h and we used the random-effect model to analysis the outcome of them. The results showed that compared to INB, PVB resulted in no significantly difference in VAS at rest at the first 2 h (Std. MD = -0. 71; 95% CI = -2. 32to 0. 91; P = 0. 39) with significant heterogeneity among the studies ($I^2$ = 86%, heterogeneity P = 0. 0009) (Fig 6).

The sensitivity analyses conducted by us found that after omitting one study by turns, significant heterogeneity was still present among the trials. Then we performed the subgroup analysis to investigate that whether the VAS at rest at the first 1 h was influenced by the

Table 1. Characteristics of included randomized controlled studies (Data presented as mean ± standard deviation).

| Study | Paravertebral block | | | | | Intercostal nerve block | | | | | surgery |
|---|---|---|---|---|---|---|---|---|---|---|---|
| | Number | Age | Body mass index | ASA (I/II/III) | Methods | Number | Age | Body mass index | ASA (I/II/III) | Methods | |
| Vlassakovc.2020 | 10 | 54±11 | 27.1 ±4.4 | (0/8/2) | PVB was given under ultrasound-guidance with 2.5mg/kg ropivacaineprior to surgery combined with intravenous and general anesthesia. | 10 | 47±15 | 28.7 ±6.1 | (1/8/1) | INB was given under ultrasound-guidance with 2.5mg/kg ropivacaine prior to surgery combined with intravenous and general anesthesia. | Breast surgery |
| Chen.2020 | 24 | 51.6 ±104 | 22.9 ±2.6 | (9/15/0) | After anesthesia induction, a single anesthesiologist performed PVB at T5-T7 levels under ultrasound guidance using 20 ml of 0.375% ropivacaine. | 24 | 58.2 ±7.8 | 23.5 ±2.4 | (9/15/0) | After anesthesia induction, a single anesthesiologist performed ICNB at T4-T9 levels under ultrasound guidance using 20 ml of 0.375% ropivacaine. | Thoracoscopic surgery |
| Mogahed.2019 | 35 | 43.54 ±10.51 | 27.77 ±1.80 | (0/24/11) | Patients received a single-shot PVB at the T4 level with 5 mL of 1% lidocaine before induction of the GA. | 35 | 43.97 ±9.29 | 27.54 ±9.74 | (0/23/12) | Patients received thoracoscopic intercostal block infiltration after induction of anaesthesia from the third to the eighth INB, in addition to intrathoracic vagal block. | Non-Intubated Uniportal Video-Assisted Thoracoscopic Surgery |
| Wu.2019 | 34 | 58.2 ±7.8 | / | (2/23/9) | Patients received a total of 0.3ml/kg of a mixture containing 0.5% ropivacaine and 1/200000 epinephrine after placement of needles for either a single thoracic PVB under ultrasound. | 32 | 56.5 ±8.3 | / | (6/20/6) | Patients received a total of 0.3ml/kg of a mixture containing 0.5% ropivacaine and 1/200000 epinephrine after placement of needles for two individual ICNBs under ultrasound. | Thoracoscopic surgery |
| Kadomatsu.2018 | 26 | 67.9 ±8.2 | 23.3 ±3.9 | / | After catheter placement, a single dose of 20ml 0.375% ropivacaine for thoracic PVB before surgery combined with intravenous and general anesthesia. | 24 | 65.4 ±11.1 | 21.9 ±3.4 | / | After catheter placement, a single dose of 10ml 0.375% ropivacaine for INB before surgery combined with intravenous and general anesthesia. | Video-assisted thoracic surgery |
| Hutchins.2018 | 23 | 62.09 ±8.85 | 27.1 ±5.26 | (1/11/11) | 3-to-5 mL of 1.5% lidocaine with epinephrine 1:200,000 was injected under ultrasound guidance at T5/6 or T6/7 level before the catheter was placed. | 25 | 59.00 ±12.43 | 28.08 ±4.94 | (1/6/18) | They injected over the surgical site and chest tube dermatomes, which consisted of 5-to-7 dermatomes (T3-9) depending on the incisions and chest tube placement. The local anesthetic used for INB was either 0.25% or 0.5% plain bupivacaine. | Video-Assisted Thoracoscopic Surgery |
| Chen.2014 | 30 | 53±8 | 23±5 | (8/12/0) | Before anesthesia induction, a anesthesiologist performed PVB at T4-T7 levels under ultrasound guidance using 20 ml of 0.375% ropivacaine. | 30 | 56±9 | 24±3 | (6/14/0) | Before induction of general anesthesia, the INB was performed from parietal pleura to intercostal nerve at five levels with 25ml of 0.375% ropivacaine. | Thoracoscopic Surgery |

(*Continued*)

**Table 1.** (Continued)

| | Paravertebral block | | | | | Intercostal nerve block | | | | | |
|---|---|---|---|---|---|---|---|---|---|---|---|
| Study | Number | Age | Body mass index | ASA (I/II/III) | Methods | Number | Age | Body mass index | ASA (I/II/III) | Methods | surgery |
| Ouerghi.2008 | 23 | 47±13 | / | (0/15/8) | During chest closure, patients received PVB with two bolus of 0.25% bupivacaïne up to 10 ml and then a catheter was placed with continuous postoperative infusion of 0.25% bupivacaïne, at a rate of 0.1ml/Kg/h. | 23 | 44±15 | / | (0/14/9) | During chest closure, patients received INB with two bolus of 0.25% bupivacaïne up to 10 ml and then a catheter was placed with continuous postoperative infusion of 0.25% bupivacaïne, at a rate of 0.1ml/Kg/h. | Pulmonary resection with elective posterolateral thoracotomy |
| Perttunen 1995 | 15 | / | / | (3/7/5) | Patients in the PVB groups received a bolus dose of 0.25% bupivacaine according to the height of the patient before wound closure and a continuous infusion of 0.25% bupivacaine 8 ml/h. | 15 | / | / | (2/12/1) | Before wound closure at the end of operation the surgeon performed intrathoracic unilateral INB (T3–7) with a total of 16 ml of 0.5% bupivacaine. | Thoracotomy |

different nationalities of subjects. In the 4 trials, subjects of 2 studies Wu (2018) and Chen (2014) [16, 19] were Chinese and they were classified in the subgroup1 showing no significantly difference in VAS at rest at the first 1 h (Std. MD = -1.02; 95% CI = -2.74 to 0.69; P = 0.24) with significant heterogeneity among the studies ($I^2$ = 99%, heterogeneity P<.00001). At the same time, the other two studies both including non-Chinese patients Kadomatsu (2018) and Perttunen (1995) [17, 21] was classified in subgroup2, which also showed no significantly difference in VAS at rest at the first 2 h in people accepting PVB (Std. MD = 0.04; 95% CI = -1. 57to 1.66; P = 0. 96) with significant heterogeneity among the studies ($I^2$ = 59%, heterogeneity P = 0. 12) (Fig 7).

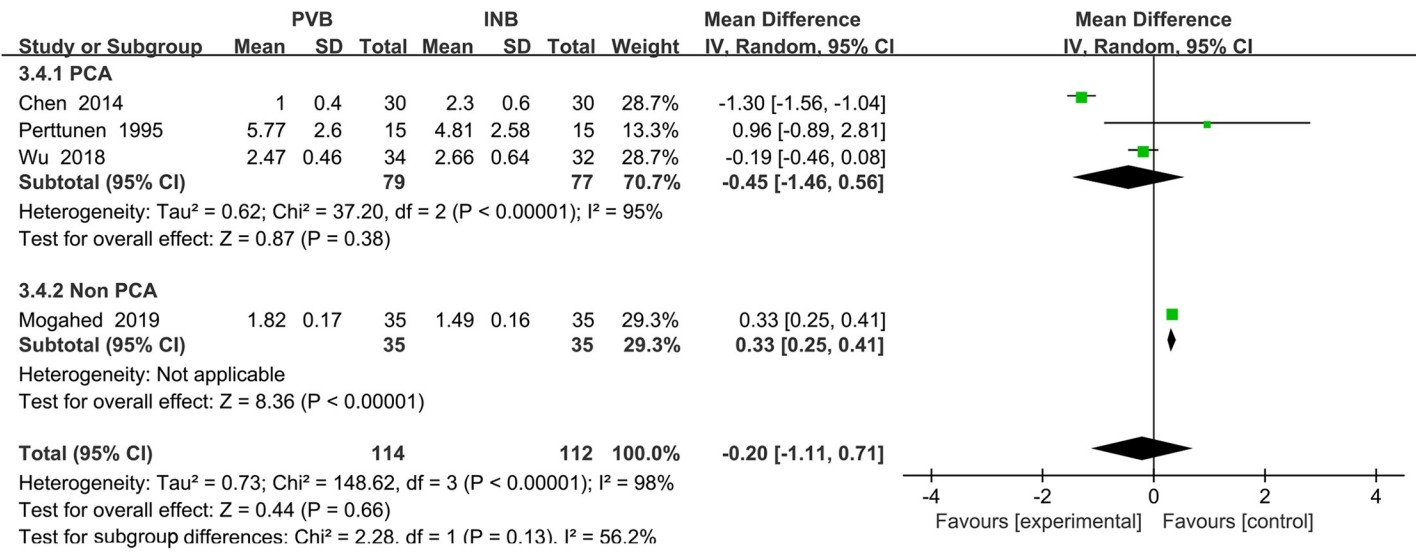

**Fig 3. Forest plot of comparison: PVB VS INB.** Outcome: VAS at rest at the first 1 h.

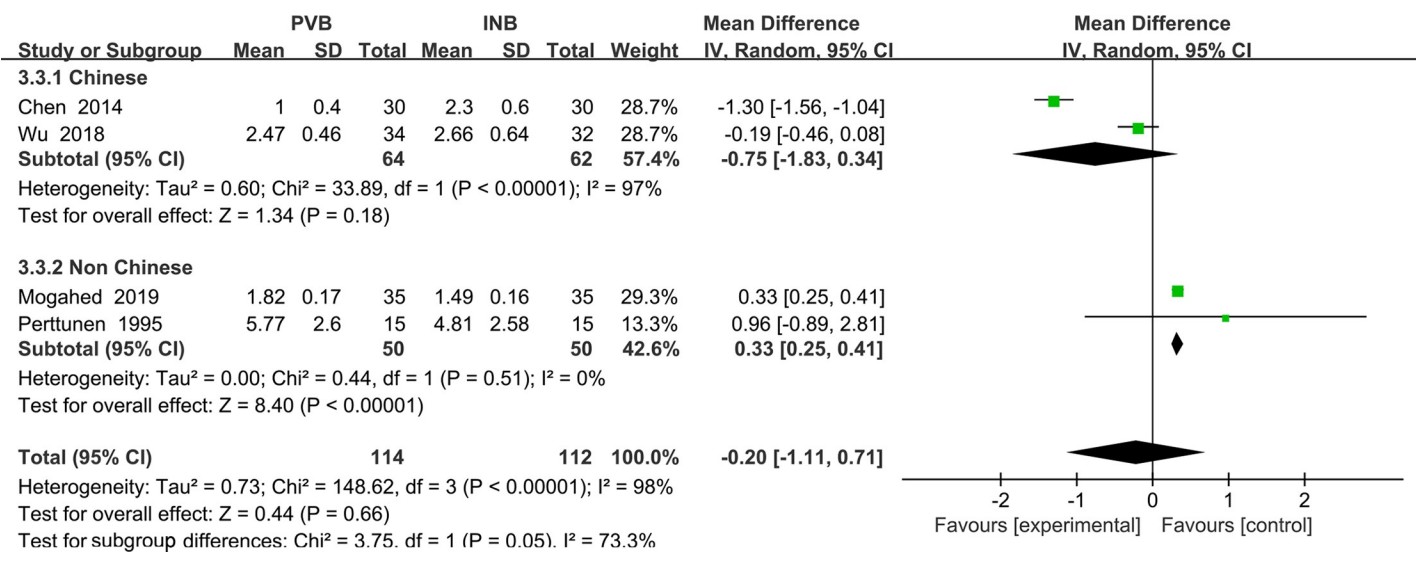

**Fig 4. Subgroup analysis of comparison: Chinese VS non Chinese.** Outcome: VAS at rest at the first 1 h.

We also performed the subgroup analysis to investigate that whether the VAS at rest at the first 2 h was influenced by the use of opioid drug or non steroidal anti-inflammatory drugs (NSAIDs). However, after excluding the trail using NSAIDs Kadomatsu (2018) [17], the heterogeneity was still very significant ($I^2$ = 97%, heterogeneity P<0. 00001) and there was no significant difference between Opioid drug subgroup and NSAIDs subgroup (Std. MD = -0.51; 95% CI = -2.00to 0.97; P = 0.50) (Fig 8).

**Primary outcome: VAS at rest at the first 24 h.** We used the random-effect model to analysis the outcome of VAS at rest at the first 24 h in four trials [16, 17, 19, 21]. The results showed that compared to INB, PVB resulted in no significantly difference in VAS at rest at 24 h (Std. MD = -0. 36; 95% CI = -0. 73to -0. 00; P = 0. 05) with significant heterogeneity among the studies ($I^2$ = 56%, heterogeneity P = 0. 08) (Fig 9).

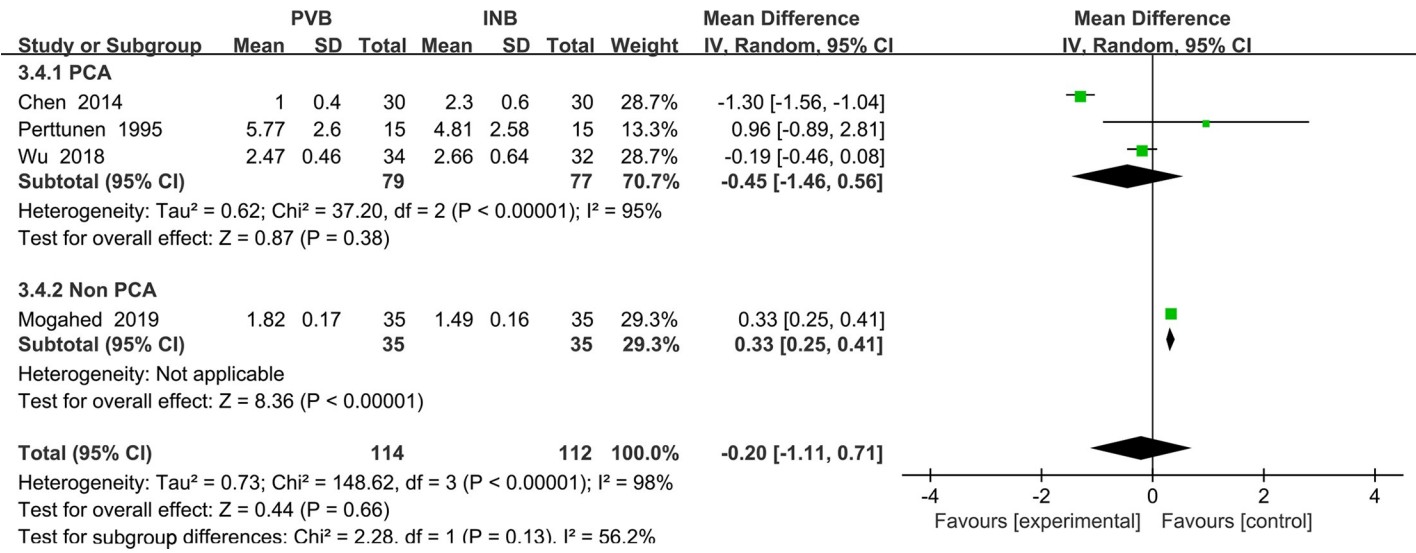

**Fig 5. Subgroup analysis of comparison: PCA VS non PCA.** Outcome: VAS at rest at the first 1 h.

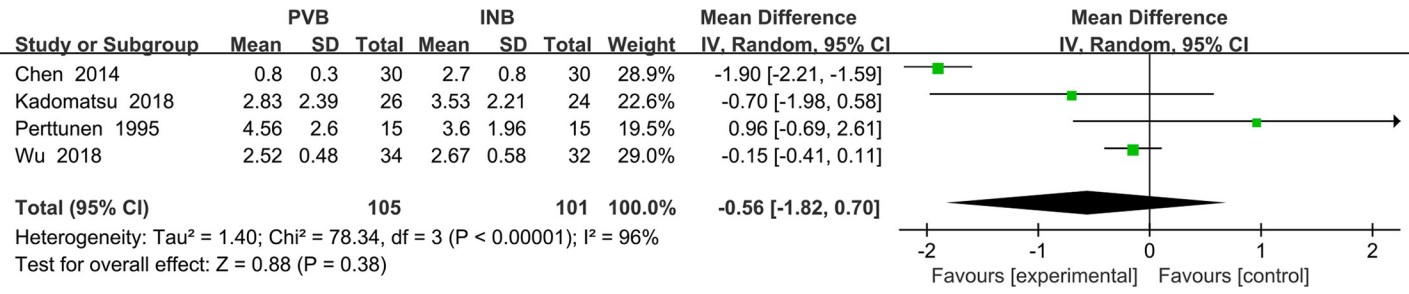

**Fig 6. Forest plot of comparison: PVB VS INB.** Outcome: VAS at rest at the first 2 h.

Because the result of P value given by Review Manage only had two decimal places, we then used Stata/MP 14.0 to analysis the outcome again and find that there is no significant difference between PVB and INB (P = 0.063) (Fig 10).

After conducting the sensitivity analysis, we found that when we excluded the study Chen (2014) [19], no heterogeneity was present among the remaining studies ($I^2$ = 0%). However, the exclusion of the study Chen (2014) [19] did not materially alter the difference VAS at rest at the 24h between PVB and INB (MD = -0. 18; 95% CI: −0. 39 to -0. 02, P = 0. 08) (Fig 11).

We performed the subgroup analysis to investigate whether the VAS at rest at the first 24 h was influenced by the different nationalities of subjects enrolled in our analysis. In the 4 trials, subjects of 2 studies Wu (2018) and Chen (2014) [16, 19] included were Chinese and they were classified in the subgroup1, which resulted in significantly decreased in VAS at rest at the first 1 h in patients accepting PVB (Std. MD = -0.32; 95% CI = -0.49 to-0.14; P = 0.0003) with significant heterogeneity among the studies ($I^2$ = 80%, heterogeneity P = 0.03). At the same time, the other two studies Kadomatsu (2018) and Perttunen (1995) [17, 21] both included non-Chinese patients and they were classified in the subgroup2, showing no significantly difference in VAS at rest at the 24h (Std. MD = -0.29; 95% CI = -1. 32 to 0.74; P = 0. 58) with significant heterogeneity among the studies. ($I^2$ = 47%, heterogeneity P = 0. 17) (Fig 12).

We also performed the subgroup analysis to investigate that whether the VAS at rest at the first 2 h was influenced by the use of opioid drug or NSAIDs. However, after excluding the

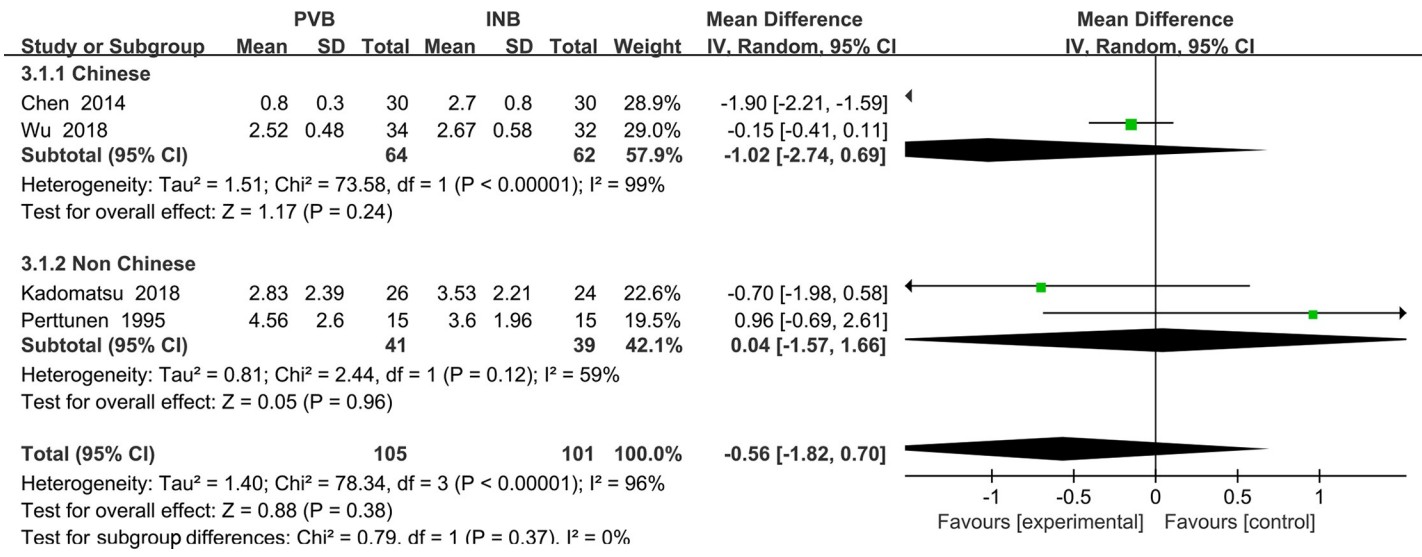

**Fig 7. Subgroup analysis of comparison: Chinese VS non Chinese.** Outcome: VAS at rest at the first 2 h.

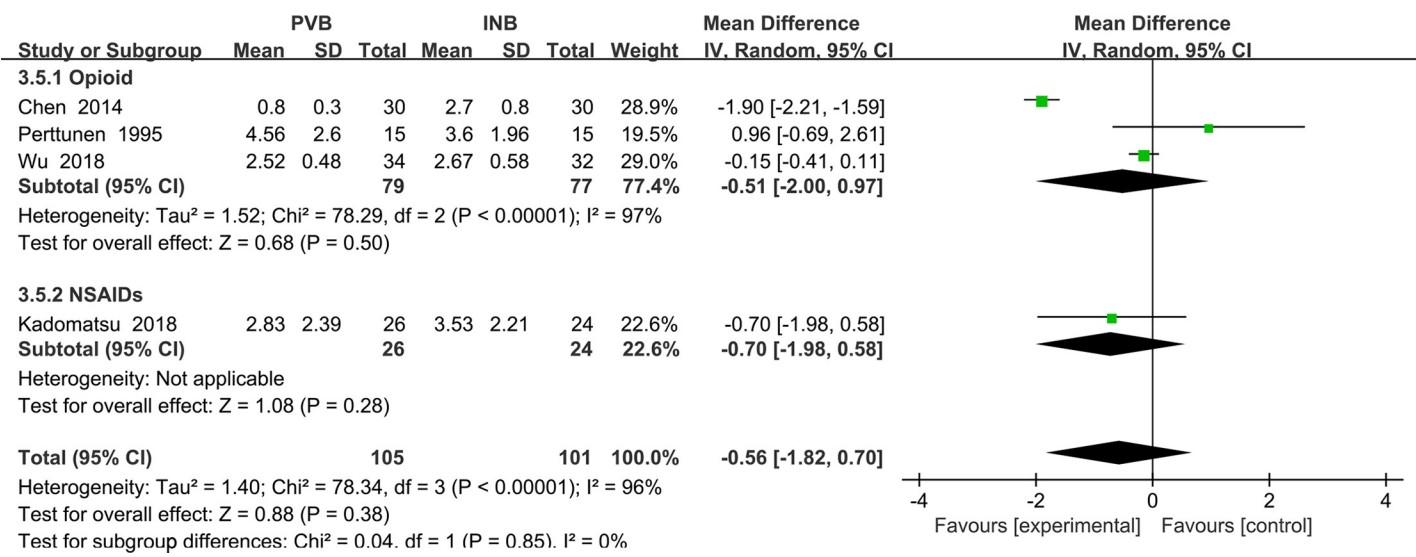

**Fig 8. Subgroup analysis of comparison: Opioids drug VS NSAIDs.** Outcome: VAS at rest at the first 2 h.

study using NSAIDs Kadomatsu (2018) [17], the heterogeneity was still very significant ($I^2$ = 67%, heterogeneity P = 0.05) and there was no significant difference between Opioid drug and NSAIDs and INB (Std. MD = -0.32; 95% CI = -0.72 to 0.08; P = 0.11) (Fig 13).

**Primary outcome: VAS at rest at the first 48 h.** We used the random-effect model to analysis the outcome of VAS at rest at 48 h. The results showed that compared to INB group, PVB resulted in no significantly difference in VAS at rest at the first 48h (Std. MD = -0. 04; 95% CI = -0. 20to 0. 11; P = 0. 57) without significant heterogeneity among the studies ($I^2$ = 0%, heterogeneity P = 0. 64) (Fig 14).

Although there was no heterogeneity in above analysis, we still performed the subgroup analysis to investigate that whether the VAS at rest at 48h was influenced by the use of opioid drug or NSAIDs. After excluding the study using NSAIDs Kadomatsu (2018) [17], there was also no heterogeneity in the subgroup ($I^2$ = 0%, heterogeneity P = 0.44) and there was no significant difference between Opioid drug and NSAIDs and INB (Std. MD = -0.05; 95% CI = -0.20 to 0.11; P = 0.56) (Fig 15).

## Secondary outcome

**Meta-analysis of rates of postoperative nausea and vomit.** The results of our analysis were shown in Fig 2. Data of PONV were given in six trials Chen (2020), Wu (2018),

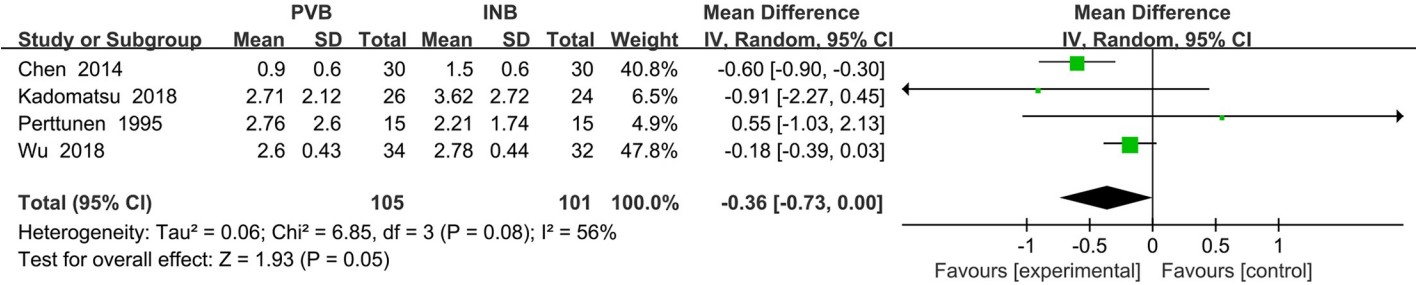

**Fig 9. Forest plot of comparison: PVB VS INB.** Outcome: VAS at rest at the first 24 h.

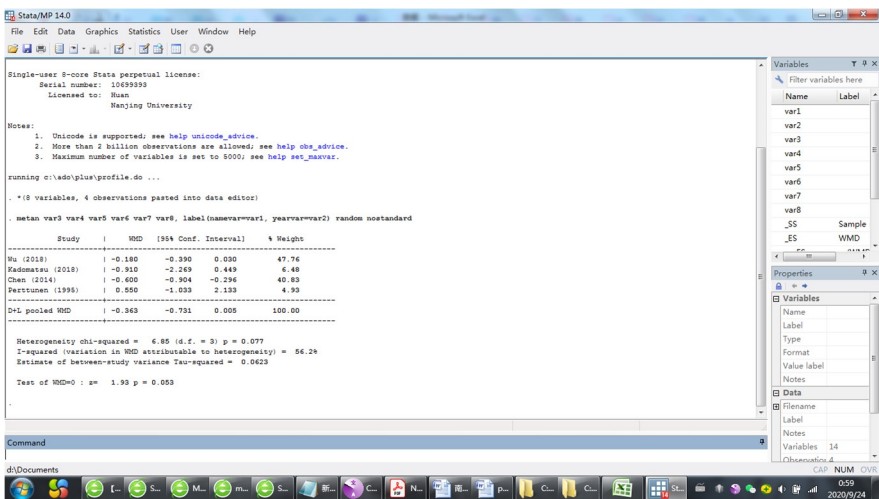

**Fig 10. Comparison: PVB vs INB.** Outcome: VAS at rest at the first 24 h.

Kadomatsu (2018), Hutchins (2017), Chen (2014) and Perttunen (1995) [14, 16–19, 21]. All in all, the rates of PONV in the PVB and INB groups were 26.5% and 36.2%, respectively. There is no significant difference in PONV between PVB and INB (OR = 0. 63; 95% CI = 0. 38 to 1. 03; P = 0. 06), and no evidence of high heterogeneity was found among the included studies ($I^2$ = 20%; P = 0. 28) (Fig 16).

**Meta-analysis of rates of postoperative additional analgesia.** The results of our analysis were shown in Fig 2. Data of postoperative additional analgesia was given in two trials Chen (2014) and Kadomatsu (2018) [14, 17]. All in all, the rates of postoperative additional analgesia in the PVB and INB groups were13. 3% and 18. 1%, respectively. There was no significant difference in rates of postoperative additional analgesia between PVB with INB (OR = 0. 57; 95% CI = 0. 21 to 1. 55; P = 0. 27), and no evidence of high heterogeneity was found among the included studies ($I^2$ = 0%; P = 0. 36) (Fig 17).

**Meta-analysis of postoperative use of morphine in 48h.** We used the random-effect model to analysis the outcome of postoperative use of morphine in three trials Hutchins (2017), Perttunen (1995) and Ouerghi (2008) [18, 20, 21]. The results showed that compared to INB, PVB can result in a significantly decrease in postoperative morphine (Std. MD = -14. 57; 95% CI = -26. 63 to -0.25; P = 0. 02), without significant heterogeneity among the studies ($I^2$ = 56%, heterogeneity P = 0. 10) (Fig 18).

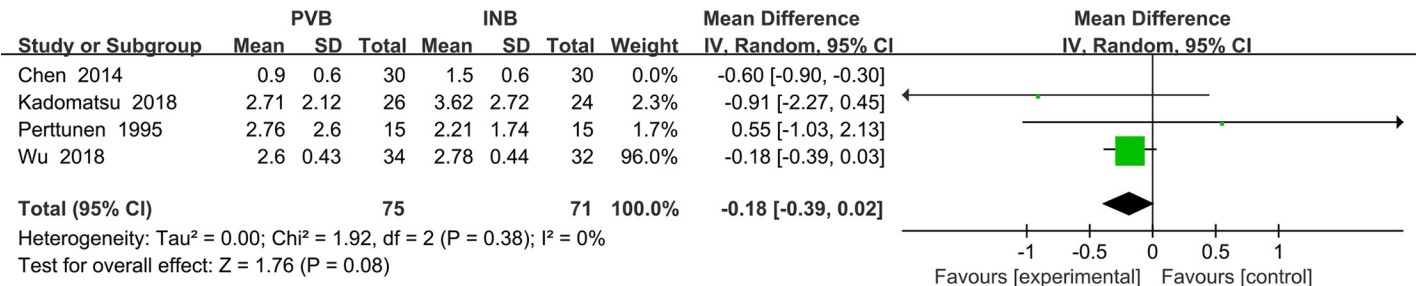

**Fig 11. Forest plot of comparison: PVB VS INB.** Outcome after exclusion of the study Chen (2014): VAS at rest at the first 24 h.

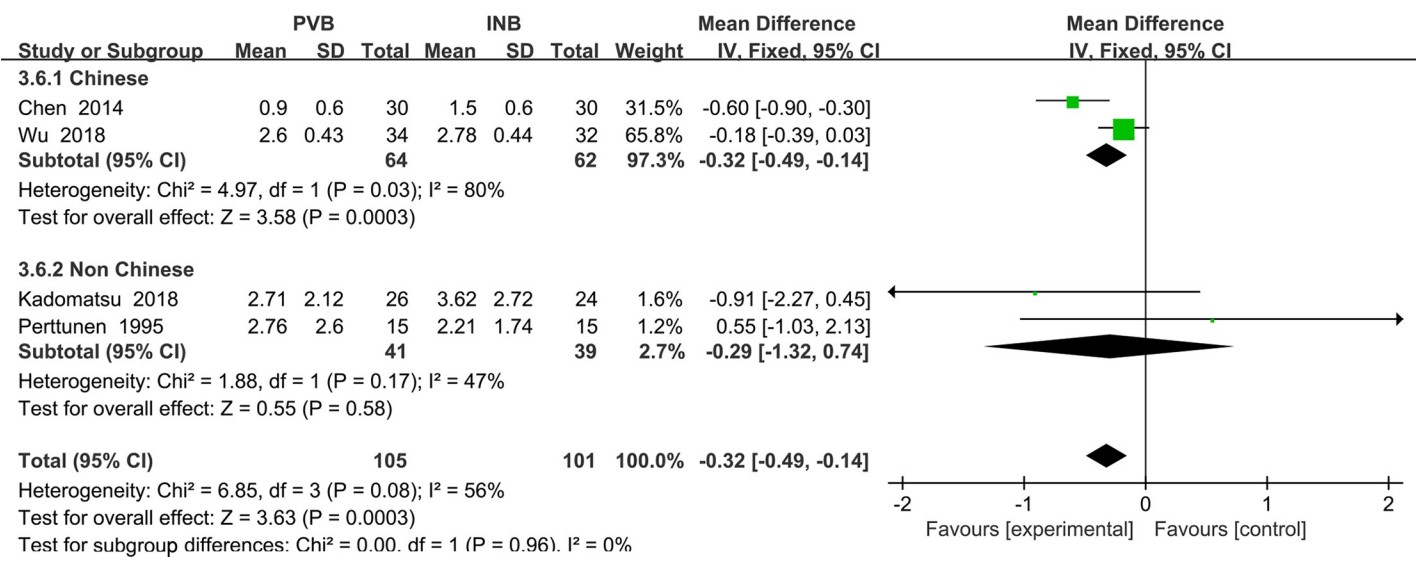

**Fig 12. Subgroup analysis of comparison: Chinese VS non Chinese.** Outcome: VAS at rest at the first 24 h.

## Discussion

### Summary of evidence

Compared with other common surgery, thoracic surgery has higher incidence of postoperative pain. In the past, thoracic epidural anesthesia was the gold standard for postoperative analgesia after thoracic surgery and breast surgery [25, 26]. However, in recent years, due to the improvement of technique such as ultrasound-guided paravertebral block, anterior serratus block, intercostal nerve block and so on, epidural analgesia has been gradually replaced [27, 28]. Many studies have shown that PVB or INB has similar analgesic effect with epidural analgesia while having less complication and better postoperative rehabilitation [29]. At the same time, they can provide satisfactory depth of anesthesia during the surgical without causing

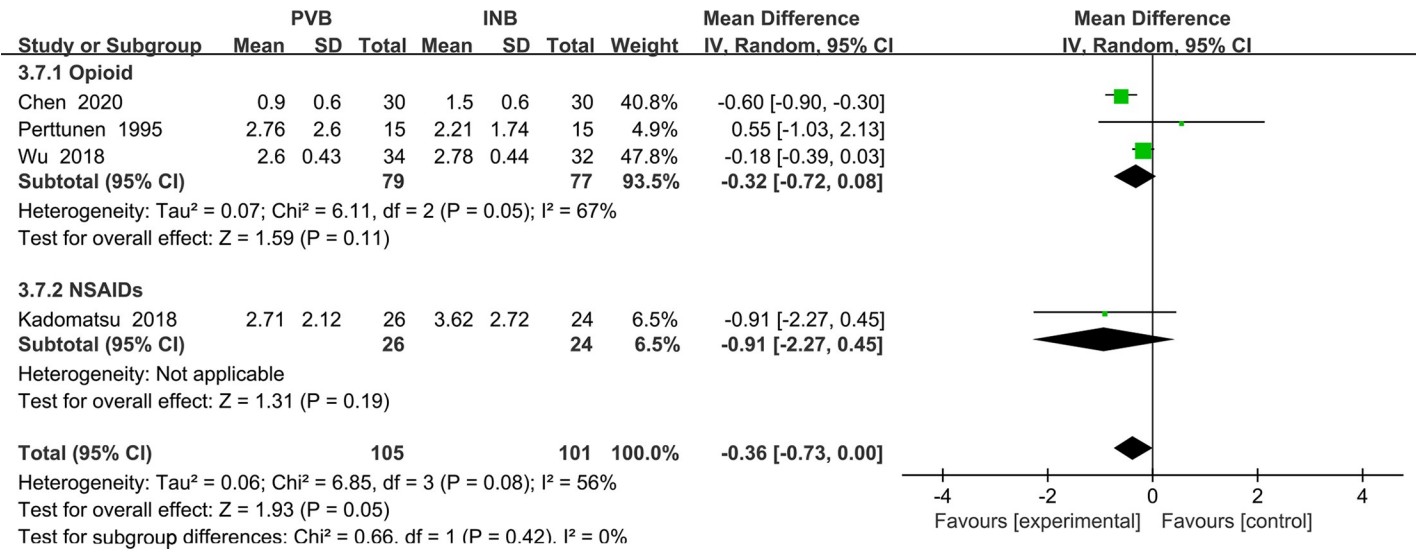

**Fig 13. Subgroup analysis of comparison: Opioids drug VS NSAIDs.** Outcome: VAS at rest at the first 24 h.

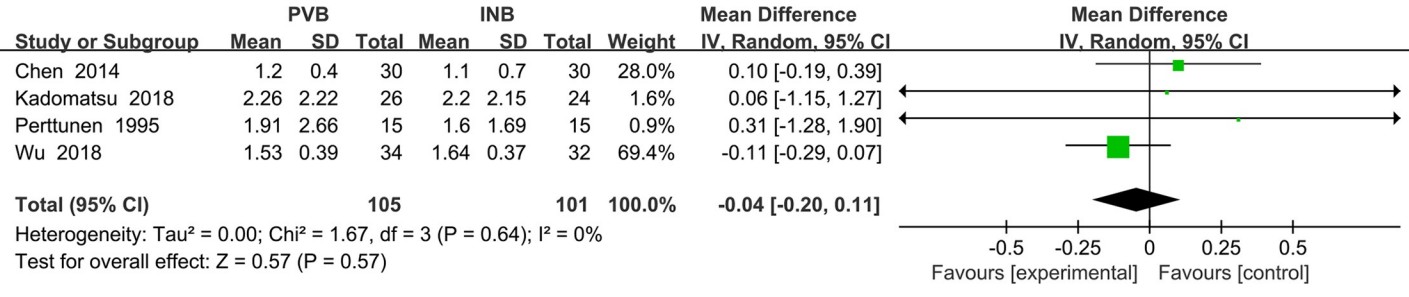

**Fig 14. Forest plot of comparison: PVB VS INB.** Outcome: VAS at rest at the first 48 h.

significant fluctuation of hemodynamic [21, 25, 30]. This meta-analysis of RCTs between PVB and INB showed that for patients undergoing thoracic surgery or breast surgery, no significant difference was found in VAS for most of time, rate of postoperative PONV and rate of postoperative additional analgesic. However, VAS of non Chinese group at 1h and VAS of Chinese group at 24h showed significant difference in our analysis, which mean PVB can provide better analgesic effect. At the same time, the consumption of morphine of PVB group in 48h is lower than that of the INB group with significant difference.

In most of the RCTs included, the authors reported postoperative pain scores, of which some were used in the form of NRS instead of VAS resulting in our relatively small sample size. The postoperative analgesia of a single PVB after thoracoscopic surgery can last from 6 to 48 hours [31], so we evaluated the VAS at 1, 2, 24, and 48 hours after surgery. In our included studies, there was a difference in the VAS at rest at 48 hours after operation reported by Kadomatsu et.al [17] and in the VAS at rest at 1hour and 2 hour after operation reported by Chen et.al [19]. In addition, some studies have also reported VAS on coughing or exercising. The VAS on coughing at 8 hours after surgery between PVB and INB reported in Chen et.al [14] had significant difference. There was also a significant difference in the VAS on coughing at 24hours and 48 hours after operation reported by Ouerghi et.al [20] and the VAS on coughing at 1hour and 2 hour after operation reported by Chen et.al [19], and the other studies included did not find any differences in the VAS on coughing in each period of time.

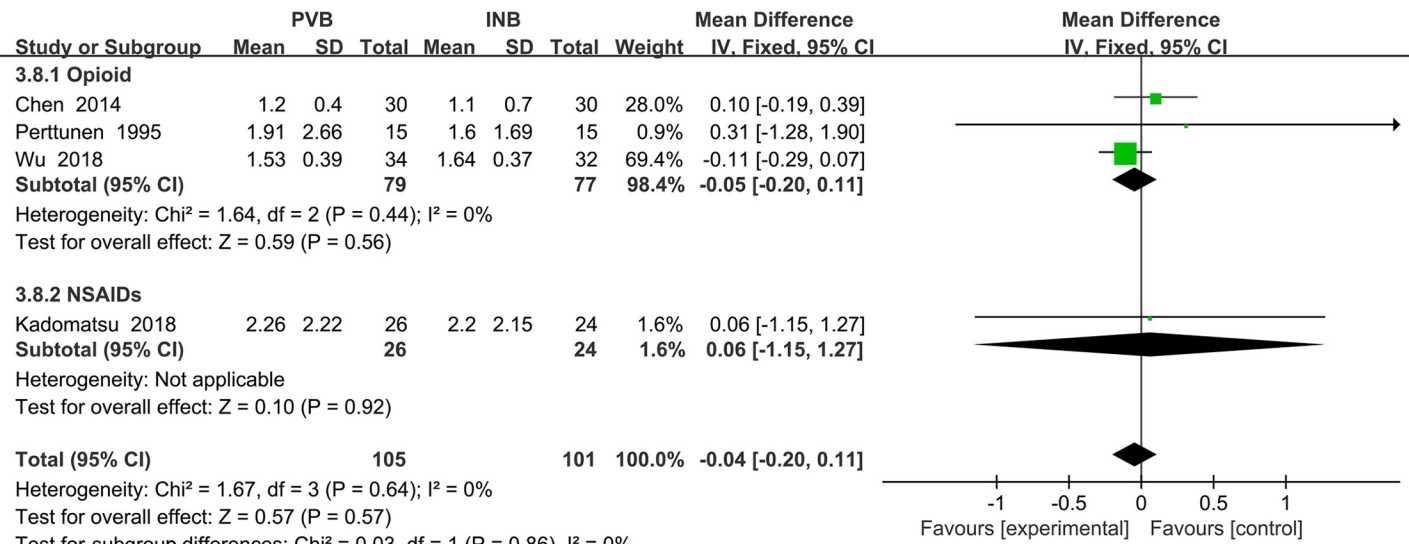

**Fig 15. Subgroup analysis of comparison: Opioids drug VS NSAIDs.** Outcome: VAS at rest at the first 48 h.

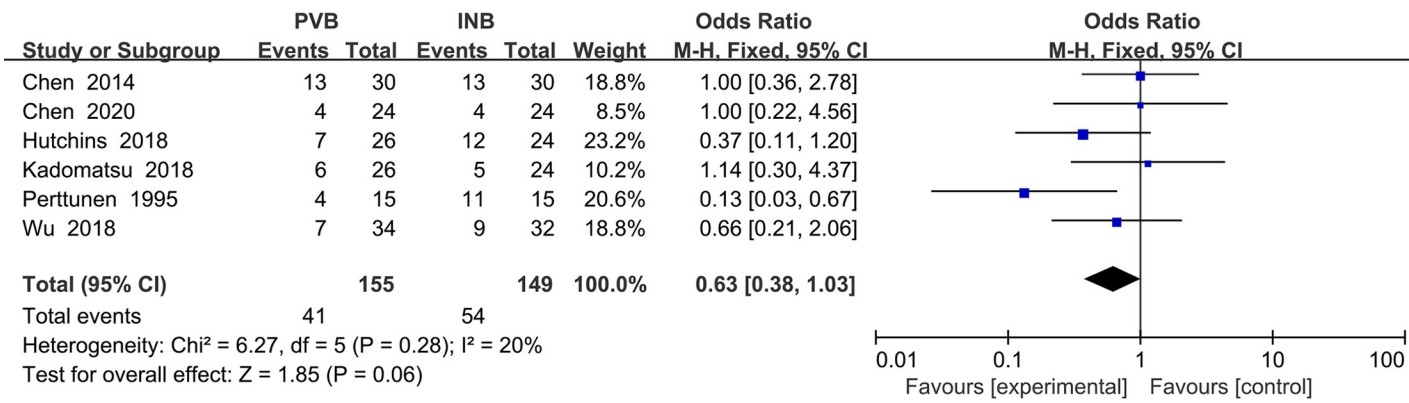

**Fig 16. Forest plot of comparison: PVB VS INB.** Outcome: Rates of postoperative nausea and vomiting.

Our meta-analysis's results were consistent with the results of some included studies proving that PVB may provide a slightly better analgesic effect, especially at 1h in non Chinese subgroup and 24h in Chinese subgroup. Although an hour after surgery is often the most painful time, Chinese patients usually lacked a clear understanding of postoperative analgesia and believed that pain is an inevitable result of surgery [32], which may be the reason of significant difference of VAS at 1h in non Chinese group. Two studies [16, 19] in VAS at 24h of Chinese subgroup both performed a single nerve block before operation rather than continuous nerve block used in non Chinese subgroup, which may lead to decreased duration of nerve block and significant differences in the analgesic effect between PVB and INB at 24 hours in Chinese subgroup.

Regarding the use of postoperative opioid, the included studies have shown contradictory conclusions. Hutchins et al. [18] found that in the interval from 24 to 48 hours, the use of morphine in the PVB group was significantly reduced compared with INB group, which may be related to the incomplete analgesic effect of INB and the narrow space around the intercostal nerves. Wu et al. [16] and Ouerghi et al. [20] also got similar conclusion. Our meta-analysis's results were consistent with the results of most included studies that compared with INB, PVB can significantly decreased the postoperative use of morphine. We have proved that the analgesic effect of PVB is better than INB, which lead to lower demand of morphine. In addition, Perttunen et.al [21] found that bupivacaine concentrations were higher of INB group than that of PVB group at 2h after the surgery while from 6 h after the surgery, bupivacaine concentrations of INB group were significantly lower than that of PVB groups. We think the rich blood flow besides intercostal nerve which led to the rapid absorption of local anesthetics may also cause the significant difference in analgesia and more use of morphine. It was worth noting

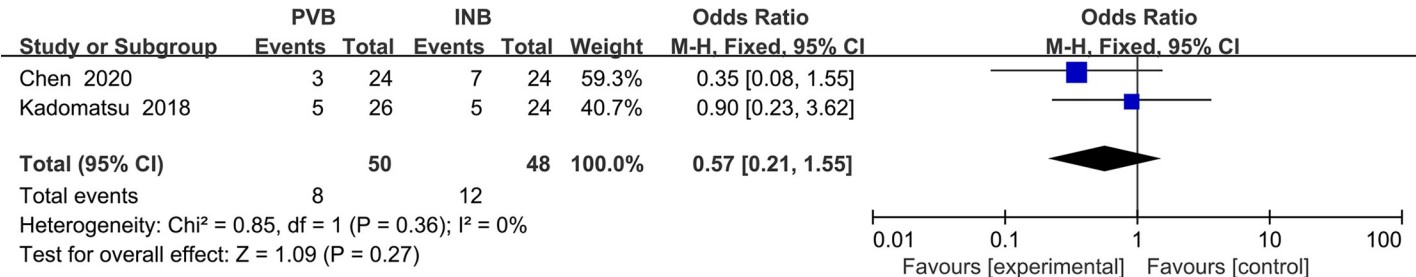

**Fig 17. Forest plot of comparison: PVB VS INB.** Outcome: Rates of postoperative nausea and vomiting.

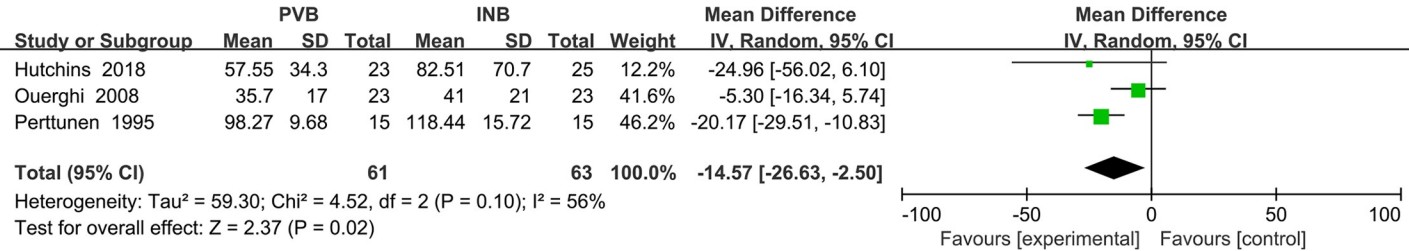

**Fig 18. Forest plot of comparison: PVB VS INB.** Outcome: Postoperative use of morphine.

that although the local anesthetic is absorbed quickly, the concentration would not exceed the threshold of poisoning under the condition of conventional dose [33].

All the included trials also showed no significant difference in the number of patients needing additional analgesics or emergency analgesics, which proved that the analgesic effect of PVB and INB meets the needs of the majority of people.

Although PVB can provide better analgesia efficacy and cause lower consumption of morphine after thoracic surgery and breast surgery. PVB had another advantage that a single injection of PVB can cover 3–6 dermatomal levels [34], while INB required multisegmental block. However, it was also necessary to be vigilant that local anesthetics in thoracic segments can spread to the neck and block the high sympathetic nerve chain, which may cause Horner syndrome.

In terms of postoperative complications, many studies have shown that the two methods have few postoperative complications. The most common complication was PONV, but there was no significant difference between the two groups. The PVB group had a slightly lower incidence of PONV than the INB group, which may be caused by a lower consumption of morphine. Nausea and vomiting were mostly caused by Vagal excitation, hypotension, stomach fullness, and use of opioids. PVB and INB did not affect the vagus and have less influence on hemodynamics than epidural anesthesia, so the incidence of vomiting was relatively lower.

This analysis did not compared the other complications in detail and only explained them as follows in the discussion because they were too rare to see in our included studies. One study found that the routine use of a single-injection, transverse, in-plane ultrasound-guided technique for PVB in patients undergoing mastectomy was associated with very few complications [35]. Our included studies have also showed that the incidence of pneumothorax, hemorrhage caused by punctured blood vessels, and local anesthetic poisoning have greatly decreased due to proper manipulation and help of ultrasound. Except for two studies [14, 21] reporting the discovery of hematomas, the rest of the studies did not even find a case of such complication. Therefore, we have to conclude that ultrasound-guided PVB and INB were very safe measures of analgesia. Interestingly, the visual score of INB has also been proven to be significantly higher than PVB [13], which may be because the intercostal nerve was shallow and the rib, pleura and muscles were well-defined. At the same time, this study also found that the length of needle insertion in the INB group was longer than that of PVB group, because the lager space between adjacent ribs allows the tip of puncture needle to reach the position better and provide better visual scoring and manipulation space.

Arrhythmia was another common complication after thoracic surgery. However, we did not conduct a meta-analysis of this indicator because there was too little data to be included. Pain may be regarded as an inducement of arrhythmia all the time. Wu [16] found that PVB group had a higher incidence of supraventricular tachycardia and atrial fibrillation than INB group. And there were significant differences between the two groups, which may be related to

the better operative analgesia of PVB. However, the causes of arrhythmia were various, and it may also be caused by increased right ventricular preload, hyperadrenergic conditions, and an imbalance of the autonomic nervous system in the form of sympathetic tone [36]. In terms of the use of β blockers after surgery, PVB group had also decreased significantly. We had reason to believe that it was a better choice to perform PVB for patients with cardiac dysfunction especially arrhythmia.

Postoperative respiratory depression was also mostly caused by pain. In addition to making patients unwilling to inhale deeply, reducing the tidal volume, and increasing respiratory rate, it may also inhibit cough reflexes. Postoperative analgesia after surgery was an effective measure to prevent respiratory depression, atelectasis, and lung infections [37, 38]. In our included studies, only one study found several cases of respiratory depression [21], and the rest of the studies did not report such implications. We thought that this condition was also caused by poor postoperative analgesia and non ultrasound-guided puncture. It was also possible that the patients' own coexisting cardiac dysfunction and increased blood concentration of ropivacaine caused hypoxic pulmonary vasoconstriction leading to respiratory depression. Except for the study performed by Ouerghi et.al [20] finding that FEV1 of PVB group was significantly higher than that of INB group at 48th hours after surgery, the difference of FEV1 between the two groups was not found at any time. One literature once reported that VAS was a related factor for postoperative FEV1 changes [37], which can explain the above-mentioned difference of FEV1.

Finally, the incidence of central nervous system complications such as dizziness, drowsiness, and delirium were very low. Pruritus and rash were rare to see and no case of urine retention or hypotension was found.

## Heterogeneity

In our meta-analysis, the heterogeneities associated with VAS at first 1h and 2h were relatively high. After performing the sensitivity analysis and subgroup analysis, heterogeneity in non Chinese subgroups decreased significantly while heterogeneity in non Chinese groups was still significant. In terms of VAS at first 12h, we performed sensitivity analysis and found that when excluding the study of Chen [19], no heterogeneity was present among the other studies. We therefore believed that heterogeneity resulted from different sensitivity of people in different countries to pain and difference between single block and continuous block.

In most of the studies, opioid drug or NSAIDs in PCA combined with nerve block were used for postoperative analgesia. Although PVB and INB were the main analgesic factors within 48 hours after surgery, we still worried that opioids or NSAIDs were the sources of heterogeneity. We therefore did some subgroup analysis to discuss this, but the results showed that whether to give painkiller or not and the type of painkiller were not the reason for the high heterogeneity of VAS after surgery.

We thought there were several reasons that could explain the previously significant heterogeneity. Firstly, the anesthesia method of Wu [16] was non-intubated anesthesia combined with general intravenous anesthesia and nerve block. The rest of the included experiments were general anesthesia with endotracheal intubation. Secondly, the concentration of anesthetic drugs was different. For example, the concentration of ropivacaine was 0. 2%, 0. 375% or 0. 5% in different included studies. In some experiments, 1/200000 adrenaline was also added to the local anesthetic. Thirdly, difference between single block and continuous block and whether to place postoperative catheters also had a significant impact on the anesthetic effect. Fourthly, the procedures of thoracic surgery were different. For example, thoracotomy surgery was more traumatic than thoracoscopy surgery and may cause more adverse reactions in

which postoperative analgesia may be more demanded. Fifthly, another possible reason was that some of the included RCTs provided the median, range, and the first and third quartiles for these outcomes. Transformation of these data to sample means and standard deviation for meta-analysis may have introduced errors, thus causing heterogeneity between studies.

## Limitation

This study has several limitations as follows. Firstly, Although we used sensitivity analysis and subgroup analysis to decrease the heterogeneity, significant heterogeneity still existed in some of the subgroups. Secondly, many included studies only presented data of nausea and vomiting but not other common complications (hemorrhage, arrhythmia, postoperative respiratory etc.). Only two studies compared rate of arrhythmia between PVB and INB. We therefore could not give a comprehensive evaluation of postoperative safety between them. Finally, the sample size of the studies we have included was smaller compared with other meta-analysis, which may weaken our conclusion. Large samples and multicenter RCTs should be performed for further discussion.

## Conclusion

In our meta-analysis, PVB and INB had no significant difference in the incidence of postoperative additional analgesia and the rate of postoperative PONV. Moreover, PVB can lead to decrease in postoperative use of morphine. In terms of analgesic effect, VAS of non Chinese subgroup at 1 h and Chinese subgroup at 24 h of PVB was significantly lower than that of INB, which means PVB can provide better postoperative analgesia than INB. In some included studies, PVB had a lower incidence of postoperative arrhythmias and a better recovery of FEV1, which can be used as a preferred method in patients with cardiopulmonary diseases in thoracic surgery or breast surgery.

On the whole, PVB was a more secure and effective method for postoperative analgesia.

## Supporting information

**S1 Appendix. Search strategy.**
(DOCX)

**S2 Appendix. Egger's test.**
(DOCX)

**S3 Appendix. PRISMA checklist.**
(DOC)

## Author Contributions

**Conceptualization:** Sheng Huan, Guoping Yin.

**Data curation:** Sheng Huan, Yihao Ji.

**Formal analysis:** Sheng Huan, Yihao Ji.

**Methodology:** Sheng Huan.

**Project administration:** Sheng Huan.

**Resources:** Sheng Huan.

**Software:** Sheng Huan, Yihao Ji.

**Supervision:** Sheng Huan, Youming Deng, Jia Wang, Guoping Yin.

**Validation:** Sheng Huan.

**Visualization:** Sheng Huan.

**Writing – original draft:** Sheng Huan.

**Writing – review & editing:** Sheng Huan, Youming Deng, Jia Wang, Guoping Yin.

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
