## [Decision Letter · Decision Letter 0]

26 Jun 2020

PONE-D-20-13360

Efficacy and safety of paravertebral block versus intercostal nerve block in Thoracic surgery or Breast surgery: a systematic review and meta-analysis

PLOS ONE

Dear Dr. Sheng Huan

Thank you for submitting your manuscript to PLOS ONE. After careful consideration, we feel that it has merit but does not fully meet PLOS ONE’s publication criteria as it currently stands. Therefore, we invite you to submit a revised version of the manuscript that addresses the points raised during the review process.

I would appreciate if you make a careful attention to the reviewer's comments specially the statistical ones. 

We look forward to receiving your revised manuscript.

Kind regards,

Ehab Farag, MD FRCA FASA

Academic Editor

PLOS ONE

Journal Requirements:

2. At this time, we ask that you please provide the full search strategy and search terms for at least one database used as Supplementary Information.

3. Please provide the results of the Egger's test used to assess publication bias in the meta-analysis as a separate Figure in the manuscript.

4. We suggest you thoroughly copyedit your manuscript for language usage, spelling, and grammar. If you do not know anyone who can help you do this, you may wish to consider employing a professional scientific editing service.  

'No'

6. Please include your tables as part of your main manuscript and remove the individual files. Please note that supplementary tables (should remain/ be uploaded) as separate "supporting information" files

Additional Editor Comments (if provided):

Reviewers' comments:

Reviewer's Responses to Questions

**Comments to the Author**

1. Is the manuscript technically sound, and do the data support the conclusions?

Reviewer #1: Partly

2. Has the statistical analysis been performed appropriately and rigorously? 

Reviewer #1: No

3. Have the authors made all data underlying the findings in their manuscript fully available?

Reviewer #1: Yes

4. Is the manuscript presented in an intelligible fashion and written in standard English?

Reviewer #1: No

5. Review Comments to the Author

Reviewer #1: The authors conducted a meta-analysis on the efficacy and safety of paravertebral block vs. intercostal nerve block in Thoracic surgery or breast surgery. A total of 10 randomized trials were included and VAS pain score were assessed as the primary outcome.

Major Issue

1. The authors included RCTs for the meta-analysis, however, ref 19, Perez Herrero (2016). presented a prospective observational study which should not be included in the analysis.

2. When assessing the efficacy of the blocks, the authors compared VAS pain scores in the primary analysis, and postoperative additional analgesia/postoperative use of morphine in the secondary outcomes. As opioid strategy greatly (additional analgesia allowed yes/no, additional analgesia standardized yes/no) affects pain scores, the authors should describe the opioid strategy of each study before comparing the pain score. The authors should also explore whether opioid strategy could be the source of heterogeneity. As an example, ref 16 reported no difference in pain score but significant difference in Sufentanil consumption while ref 18 reported no difference in both pain score and opioid consumption within the first 24 hours.

Minor issues

1. The authors included 9 studies from 2008-2020 and 1 study from 1995. One of the concerns is whether there are improvements in the technique of the blocks that make the 1995 one less comparable and less valuable as a reference.

2. For all the forest plots, Experimental vs. Control is very confusing. Please directly label PVB, INB.

3. The authors should check for spelling and grammar errors. For example “trials” was misspelled as “trails”, “Can not” should be spelled as “cannot”.

4. When p-values is exactly 0.05, report P-values in 3 decimal points to show significance/nonsignificance.

5. When referring to trials in the result, use author name and year of the trial instead of reference number. It’s hard to know what trial [20] is without pulling down to the reference list.

6. PLOS authors have the option to publish the peer review history of their article (what does this mean?). If published, this will include your full peer review and any attached files.

Reviewer #1: No

---

## [Author Response · Author response to Decision Letter 0]

20 Jul 2020

Dear Editors:

I am very grateful to you for giving me an opportunity of major revision, and I also cherish this chance. In view of your questions, my team and I have made a lot of efforts to improve our articles to meet your requirements during the past few days. The following is the reply to your suggestions.

Response to Journal Requirements:

1.We have tried our best to revise the manuscript to meet PLOS ONE's style requirements, including those for file naming. However, we hope the editor could review it again because we can't guarantee that the revised manuscript will fully meet the requirements of the journal.

We didn't find the specific font requirements, so we changed the font format of the article to Times New Roman. Due to many changes in the format, we did not show it in the revised manuscript with track changes. For the format of the figures, we refer to the previous articles and modify them. All the figures in our analysis are inserted into the manuscript. In addition, we also convert these images to TIF format and upload them separately in a separate folder.

We put the table of characteristics, research strategy, PRISMA checklist and results of egger's test at the end of the article in the form of supporting information because they take up too much space in the manuscript and make it look bloated. 

We use Endnote to list references and try our best to make them conform to the standard. If we have meet your standard in these terms, please point out the errors and we will correct them in the subsequent revision.

There are so many small mistakes in our original manuscript that we can't express them all in the Revised Manuscript with Track Changes, so please forgive us that we haven't modified and marked on it too much.

2.We have provided the search strategy and search terms for database used as Supplementary Information 2. We do not have other researching tools, so we can not fully retrieve some unpublished literature.

3.We've performed the published bias test and the results of the Egger's test used to assess publication bias in the meta-analysis as Supplementary Information 3.

4.We have copyedited our manuscript for language usage, spelling, and grammar again and again with the help of two chief doctors. We didn't give the article to AJE directly for polishing, because I was just a graduate student and didn't have a fixed income. If the editor thinks that there are no missing items in the method and conclusion of my revised article, but there are still some mistakes in writing, we can ask AJE company to polish the revised manuscript before we submit our manuscript next time.

5.We can confirm that The authors received no specific funding for this work. In fact, all the expenses of the article, including the layout fee, are borne by me personally.

Response to Major Issue

1. By reviewing the literature we included, we confirmed that Perez Herrero (2016) was a prospective study and does not meet the criteria of randomized controlled trials. We have therefore eliminated Perez Herrero (2016), and revised the number of included documents, total number of people, reference number, flow chart, Risk of bias graph and results of meta-analysis. But we can't show this in the Revised Manuscript with Track Changes, so we only show it in the new manuscript.

2.As for what you said that opioids used in PCA may affect the VAS of patients after operation, I have several different opinions. 

First, because of the severe trauma of thoracic surgery, PCA is the choice of analgesia in almost all the included studies. Most of the studies we included compared the analgesic effects of PCA plus PVB and PCA plus INB, which also confirmed my view that the use of PCA and opioids was inevitable. We have reasons to think that PCA is the source of heterogeneity, but we can not believe that PCA affects the reliability of experimental results and conclusions. If the use of opioids after surgery affects the comparison of analgesic effects between PVB and INB, all similar studies comparing the analgesic effect of measures of nerve block will all be considered to be unreliable because they do not control the variables, that is, the addition of opioids after operation. The object of clinical study is patients rather than animals, so we can't control all the other variables and only compare the analgesic effect of PVB and INB.

Second, I reviewed the relevant literature【1-3,5,7,8】finding that compared with simple administration of postoperative opioids or NSAIDs, the pain of patients with additional PVB or INB is significantly relieved, especially in these thoracic surgery or breast surgery. We have reasons to believe that PVB and INB are the more important analgesic factors than opioids after operation. These literatures have proved that at different measurement time nerve block plus PCA can provide a better analgesia. One of these studies【1】even found that single-dose PVB combined with PCA still had better analgesic effect even at 72 hours after surgery, not to mention the analgesic effect of continuous PVB combined with PCA. If PCA is the main analgesic factor, PCA's analgesic efficacy would be enough to cover nerve block so that there would be no difference in the results of the literature mentioned above, which is obviously inconsistent with the actual conclusion. As mentioned in one study【6】, the efficacy of epidural anesthesia is better than that of INB at the level of and two levels above and below the thoracotomy, so epidural analgesia combined with INB has no significant improvement on postoperative pain. In fact, nerve block alone is enough for most people. A study 【4】found that PVB alone has better analgesic effect than PCA. Its pain scores of PVB group is all lower than 2.0.

Third, we conducted a subgroup analysis to discuss whether the use of opioids after surgery might be the source of heterogeneity, but the conclusion shows that opioids are not the cause of high heterogeneity of VAS. On the contrary, we found that the heterogeneity may come from other aspects. Differences in pain tolerance of people in different countries and differences between single block and continuous block may be the source of heterogeneity.

In summary, we believe that VAS after surgery is the result of the joint action of nerve block and PCA, and we do not think that opioids or non steroidal anti-inflammatory drugs affect the effect of nerve block. In fact, of the five studies finally included in the VAS discussion, three used opioids after surgery, one used non steroidal anti-inflammatory drugs, and one did not indicate whether to use analgesics. We can talk about whether nerve block can reduce the use of these painkillers or whether the use of opioids after surgery might be the source of heterogeneity, but we can't think that the experiment didn't control the variables of analgesia because of the combination of painkillers.

Response to Minor Issue

1.We reviewed the article of perttunen K (1995). In their research, the experimental design and specific operation were very rigorous. As far as PVB and INB were concerned, these two techniques have not developed much since 1995 except that they are performed more accurately under the guidance of ultrasound. At the same time, we have noticed that PVB is performed by surgeons placing catheters beside the sympathetic trunk before the end of thoracotomy, with the accuracy even better than that under the guidance of ultrasound; and INB is also performed by surgeons with local anesthetics before the intercostal incision is closed, so there is no doubt about its accuracy.

2.We have replaced the “experience” and “control” in the forest plot with the label “PVB” and “INB”, which is more intuitive and clear, and easy for editors and readers to read.

3.This article has been repeatedly checked and copyedited by us to ensure that errors of language usage , grammatical and spelling are kept to a minimum. In addition, some medical proper terms may be out of the way which were gotten by looking up dictionaries. If there are some mistakes that we haven't found, please inform us directly without hesitation. We will try our best to correct them before the next submission.

4.We cannot give a result with a p value in 3 decimal points with Review Manage because the result of P value given by it only has two decimal points, we then use Stata/MP 14.0 to analysis the outcome again and find that there is no significant difference between PVB and INB (P=0.063).

5.You suggested that we use the names and years of the authors rather than the serial numbers of the references in the result. Although this is helpful for reading, I checked some meta-analysis literatures published by PLOS One during recent years and found that no other meta-analysis did so. I changed the format of my article according to your requirements, but I'm not sure if this will compound the format requirements of PLOS One journal. I hope you can consult with the editor and give us the response.

Finally, I would like to extend my sincere wishes to the editor and the reviewer for a healthy life and a smooth career.

1. Li XL, Zhang Y, Dai T, Wan L, Ding GN. The effects of preoperative single-dose thoracic paravertebral block on acute and chronic pain after thoracotomy: A randomized, controlled, double-blind trial. Medicine. 2018;97(24):e11181. Epub 2018/06/15. doi: 10.1097/md.0000000000011181. PubMed PMID: 29901652; PubMed Central PMCID: PMCPMC6023649.

2. Vogt A, Stieger DS, Theurillat C, Curatolo M. Single-injection thoracic paravertebral block for postoperative pain treatment after thoracoscopic surgery. British journal of anaesthesia. 2005;95(6):816-21. Epub 2005/10/04. doi: 10.1093/bja/aei250. PubMed PMID: 16199417.

3. Gacio MF, Lousame AM, Pereira S, Castro C, Santos J. Paravertebral block for management of acute postoperative pain and intercostobrachial neuralgia in major breast surgery. Brazilian journal of anesthesiology (Elsevier). 2016;66(5):475-84. Epub 2016/09/04. doi: 10.1016/j.bjane.2015.02.007. PubMed PMID: 27591461.

4. Fallatah S, Mousa WF. Multiple levels paravertebral block versus morphine patient-controlled analgesia for postoperative analgesia following breast cancer surgery with unilateral lumpectomy, and axillary lymph nodes dissection. Saudi journal of anaesthesia. 2016;10(1):13-7. Epub 2016/03/10. doi: 10.4103/1658-354x.169468. PubMed PMID: 26955304; PubMed Central PMCID: PMCPMC4760034.

5. Fortier S, Hanna HA, Bernard A, Girard C. Comparison between systemic analgesia, continuous wound catheter analgesia and continuous thoracic paravertebral block: a randomised, controlled trial of postthoracotomy pain management. European journal of anaesthesiology. 2012;29(11):524-30. Epub 2012/08/24. doi: 10.1097/EJA.0b013e328357e5a1. PubMed PMID: 22914044.

6. Ranganathan P, Tadvi A, Jiwnani S, Karimundackal G, Pramesh CS. A randomised evaluation of intercostal block as an adjunct to epidural analgesia for post-thoracotomy pain. Indian journal of anaesthesia. 2020;64(4):280-5. Epub 2020/06/04. doi: 10.4103/ija.IJA_714_19. PubMed PMID: 32489201; PubMed Central PMCID: PMCPMC7259421.

7. Lukosiene L, Rugyte DC, Macas A, Kalibatiene L, Malcius D, Barauskas V. Postoperative pain management in pediatric patients undergoing minimally invasive repair of pectus excavatum: the role of intercostal block. Journal of pediatric surgery. 2013;48(12):2425-30. Epub 2013/12/10. doi: 10.1016/j.jpedsurg.2013.08.016. PubMed PMID: 24314181.

8. Barr AM, Tutungi E, Almeida AA. Parasternal intercostal block with ropivacaine for pain management after cardiac surgery: a double-blind, randomized, controlled trial. Journal of cardiothoracic and vascular anesthesia. 2007;21(4):547-53. Epub 2007/08/07. doi: 10.1053/j.jvca.2006.09.003. PubMed PMID: 17678782.

 Yours sincerely

 Sheng Huan

---

## [Editor Report · Decision Letter 1]

24 Jul 2020

Efficacy and safety of paravertebral block versus intercostal nerve block in Thoracic surgery and Breast surgery : A systematic review and meta-analysis

PONE-D-20-13360R1

Dear Dr. Huan,

We’re pleased to inform you that your manuscript has been judged scientifically suitable for publication and will be formally accepted for publication once it meets all outstanding technical requirements.

Kind regards,

Ehab Farag, MD FRCA FASA

Academic Editor

PLOS ONE

Additional Editor Comments (optional):

Reviewers' comments:

Dear Dr. Shen Huan Thank you for submitting the revised manuscript. It is my pleasure to accept the manuscript for publication in PLOS ONE. Sincerely Ehab Farag MD FRCA FASA

---

## [Editor Report · Acceptance letter]

31 Aug 2020

PONE-D-20-13360R1 

Efficacy and safety of paravertebral block versus intercostal nerve block in Thoracic surgery and Breast surgery : A systematic review and meta-analysis 

Dear Dr. Huan:

I'm pleased to inform you that your manuscript has been deemed suitable for publication in PLOS ONE. Congratulations! Your manuscript is now with our production department. 

Kind regards, 

on behalf of

Dr. Ehab Farag 

Academic Editor

PLOS ONE